# Early Childhood Caries in 3 to 5 Year Old Children in Trinidad and Tobago

**DOI:** 10.3390/dj7010016

**Published:** 2019-02-07

**Authors:** Tricia Percival, Julien Edwards, Salvacion Barclay, Bidyadhar Sa, Md Anwarul Azim Majumder

**Affiliations:** 1School of Dentistry, Faculty of Medical Sciences, University of the West Indies, St. Augustine Campus, Trinidad; jdarryledwards@gmail.com (J.E.); Vacion77@gmail.com (S.B.); Bidyadhar.Sa@sta.uwi.edu (B.S.); 2Faculty of Medical Sciences, University of the West Indies, Cave Hill Campus BB11000, Barbados; azim.majumder@cavehill.uwi.edu

**Keywords:** early childhood caries, preschool children, oral health, Trinidad and Tobago

## Abstract

Background: This study was done to evaluate the prevalence and contributory factors of early childhood caries (ECC) and severe ECC (S-ECC) among preschool children of Kindergartens and Early Childhood Centres in Trinidad and Tobago. Methods: A cross-sectional study was carried out involving 342 preschool children aged 3 to 5 years. The school staff distributed a structured questionnaire to the children to be completed by the mother. Clinical examinations were conducted by calibrated examiners. Statistical significance was set at *p* < 0.05 in all analyses. Results: The prevalence of ECC and S-ECC was 50.3% and 52.3%, respectively. Dietary and oral hygiene factors particularly with bottle feeding and high plaque levels were major contributors to dental caries in this population. Conclusion: ECC and S-ECC are significant issues that occur in preschool children in Trinidad and Tobago. The development of ECC and S-ECC can be attributed to certain environmental factors like dietary habits and oral hygiene practices. Early dental assessment, broad-based oral health education programmes, increased parental/guardian engagement during oral hygiene practices and greater access to facilities for early childhood caries prevention and management can help alleviate the problems of ECC and S-ECC in this population.

## 1. Introduction

According to the American Academy of Paediatric Dentistry (AAPD), the presence of one or more decayed teeth, missing (due to caries) or filled tooth surfaces in any primary tooth in a child aged 71 months or younger is considered to be Early Childhood Caries (ECC) [1].

The definition of severe early childhood caries (S-ECC) is any sign of smooth-surface caries in a child younger than three years of age and from ages three through five, of one or more cavitated, missing (due to caries) or filled smooth surfaces in primary maxillary anterior teeth or a decayed, missing or filled score of greater than or equal to four (age 3), greater than or equal to five (age 4), or greater than or equal to six (age 5) [2].

ECC is a multifactorial and progressive disease that is heavily influenced by particular dietary and oral hygiene practices in the presence of cariogenic microorganisms over time [3,4]. It is known to be a significant health problem in both developing and industrialised countries [5,6]. It is one of the most chronic childhood conditions and has debilitating outcomes for both the patients and their families. These outcomes include complications to children’s health such as pain, abscesses, loss of appetite and inability to eat, as well as significant economic burden to families and the health care system [6].

The prevalence of early childhood caries worldwide has been shown to be quite variable and ranges from 3 to 85% with strong correlation to economic status and ethnicity. Some developed countries, e.g., Scotland, through preventive programmes, have demonstrated some reduction in caries prevalence [7]. Dental caries in the developing world however, has become a significant public health issue and is on the rise because of the easy availability of refined sugars, compromised primary level of prevention, very few community-based programs to eradicate the disease and the lack of public knowledge about the disease [4,5,8,9,10,11].

There is limited data available on dental caries prevalence in children less than 6 years in the Caribbean. The island of Puerto Rico reported 37.4% of children aged 6–47 months to be free of decalcification lesions and frank decay [12]. Data from English-speaking islands of Antigua indicated a caries prevalence of 23% in a group of 3–4-year-olds [13], Anguilla (21%) in 24–71 months old [14] and in an educational district in central Trinidad −29% in 3–5-year-olds [15].

The islands of Trinidad and Tobago are located at the most south-easterly end of the Caribbean Sea. Its population of 1.3 million is considered to have a multi-ethnic and multicultural society which has been influenced by colonisation. According to the 2011 Population and Census demographic report, this twin island state comprises of approximately 38% East Indian descent, 36% African descent, 24% Mixed descent (including Mixed African and East Indian) and Other (Caucasian, Chinese, Syrian/Lebanese, unspecified) [16].

Significant numbers of very young patients seek emergency dental treatment related to ECC [17]. The potential for ECC to impact on the development of the permanent dentition is also of concern as a previous national oral health survey revealed high caries experience in 6–8 year olds [18]. There are no outcome studies to date and there is therefore a need for greater focus of early assessment and prevention.

The objectives of this study were to
determine the prevalence of early childhood caries (ECC) and severe ECC (S-ECC) among preschool children in Trinidad and Tobago andidentify the associated factors that contribute to or impact Early Childhood Caries in this population.

## 2. Materials and Methods

A cross-sectional oral health survey was done in the twin island state of Trinidad Tobago between January 2014 and December 2014. The country was divided into five regions—Tobago, North, Central, East and South Trinidad. The regions were determined by the county boundaries and included a mix of urban and rural areas with variations in socioeconomic status.

The sample population was obtained from a list of over 350 private and public kindergartens and Early Childhood institutions registered with the Ministry of Education. The list was divided by region and schools were selected using a lottery system. The data sample was obtained from healthy male and female children between the ages of 3–5 years attending 15 selected preschools and early childhood centres.

The sample size was obtained by using the formula by Daniel (1999) *n* = *Z*^2^*P* (1 − *P*)/*d*^2^ (with *n* = sample size; *Z* = *Z* statistic for a level of confidence; *P* = expected prevalence or proportion; and *d* = precision) [19]. The expected prevalence using data from Vignarajah (1992) [13] and a preliminary assessment of cases presenting to the Child Dental Health Unit Emergency Department, University of the West Indies, School of Dentistry is 25%. For the level of confidence of 95%, which is conventional, the *Z* value is 1.96 and *d* = 0.05. The sample size was therefore determined to be 288 patients and to cater for a refusal/failed assessment rate of 15%; more than 330 children were assessed.

Ethical approval for the study including the consent procedure was obtained from the University of the West Indies, St Augustine Campus Research and Ethics Committee.

The principals and staff members of the schools were then briefed about the research project via telephone or personal visits and supplemented with pamphlets. The school staff distributed a structured questionnaire to the children to be completed by the mother. The questionnaire included questions about the early and current dietary practices, oral hygiene regimen, oral habits as well as maternal demographics, dental experiences and knowledge of dental health. Only children with completed questionnaires and a signed consent form or with verbal consent from the mother present on the day of examination were assessed.

The clinical examinations were conducted at the kindergartens and early childhood centres by two calibrated dentists (TP and JE). The examination involved visual and tactile assessment of the dentition and soft tissues using a disposable mouth mirror, blunt probe, torch light and tongue spatula. Oral hygiene status was assessed using a modification of the simplified oral hygiene (OHI-S) index (1973) [11]. The facial and lingual surfaces of the following index teeth (51, 55, 65, 71, 75 and 85) were examined. The debris scores were added and divided by the number of surfaces assessed. Oral hygiene was classified as good when the debris scores ranged from 0.0 to 1.0 and poor when scores greater than 1.0. Sterile gauze was then used to dry the teeth. Teeth were scored for caries using the WHO (1979) [4] criteria for decayed, missing, or filled teeth DMFT index and noncavitated or early white spot enamel lesions were also recorded.

After checking for errors and making the necessary corrections, Microsoft Excel was used to create a database prior to exporting to Statistical Package for the Social Sciences (SPSS) version 21.0 (IBM Corporation, Armonk, NY, USA) software for statistical (quantitative) data analysis. Both descriptive and inferential statistics methods were used for data analysis and all hypotheses were tested at the 5% level of significance. Descriptive methods included forming frequency and percentage distribution tables and summary statistics (point estimates of single proportions and of means with corresponding standard deviations). Inferential methods included hypothesis testing (chi-squared tests of association and test of equality of proportions using Fisher’s Exact test). Binary logistic regression was used to identify predictors of ECC and S-ECC. Adjusted Odds ratios (OR) were also obtained. Further Kappa coefficients were established to determine intra- and inter-examiners reliability.

## 3. Results

Intra-examiner and inter-examiner calibration were carried out on a separate group of twenty (20) preschool children who attended the University of the West Indies Dental School, Child Dental Health Emergency clinic. The Kappa coefficient for intra-examiner and inter-examiner reliability was 0.91 and 0.89, respectively.

### 3.1. Demographics

Three-hundred-and-seventy-five questionnaires and consent forms were distributed to the parents of the children attending the selected kindergartens and early childhood centres. Twenty-two children who submitted returned questionnaires and consent forms were absent on the day of examination, six children returned consent forms only but no questionnaires and five children refused examination. The response rate of 91.2% was well above the acceptable standard for questionnaire-based surveys [20].

A total of 342 children were examined, 174 (50.9%) males and 168 (49.1%) females, aged 36 to 67 months (mean age = 48.6; SD = 8.7 months). Approximately 22% of the children examined were of East Indian descent while equally 38.5% of the children examined were of African descent and of mixed descent (parent reported). Three children were either of Caucasian or Chinese descent. The ethnic composition of the sample was an adequate reflection of the general population [16].

Of the 342 mothers, 19.8% of mothers completed primary school education, 58.9% had a secondary education or vocational training and 21.3% of mothers had university or tertiary education.

### 3.2. Prevalence of Caries

The prevalence of children with early childhood caries was 50.3% and of these 52.3% had severe ECC (S-ECC). The mean number of decayed, missing, filled teeth (DMFT) was 2.83 ± 4.1 overall and 5.63 ± 4.1 among children with caries. The caries prevalence showed similar values across the various regions of Trinidad and Tobago. There was an association between region and S-ECC. Specifically there was a significant difference between the prevalence of S-ECC among children in the East region (13.1%) and that of children in the four non-East regions (29.2%) but no differences in prevalence among the four non-East regions.

Using the WHO criteria, 49.7% (*n* = 170) were classified as being caries-free and among them 10% (*n* = 17) had noncavitated or white spot lesions. Only 4.7% (*n* = 8) of children with caries had restored teeth. Severe ECC was detected in 28.2% (*n* = 46) of 3-year-olds, 23.9 % (*n* = 33) of 4-year-olds and 26.8% (*n* = 11) of 5-year-olds.

There were no associations between caries experience and ethnicity of the child. There was however a greater percentage of males than females with caries (*p* = 0.040) (Table 1).

### 3.3. Maternal Factors and Dental Exposure

Of the 342 mothers, 18.9% visited the dentist during pregnancy. The most common reasons were for check-up and toothache. Other reasons included bleeding gums, abscesses and mobile (‘shaking’) teeth. There was no statistically significant association noted between the mother’s level of education and a child’s caries experience.

A small proportion (31.3% (107)) of children was reported to have visited the dentist or dental nurse. The mean age of the first dental visit was 34.89 months (SD = 9.6), with only 1.8% (2) children visiting by their first birthday. The majority of children (59.8%) (64/107), who visited a dental professional did so for routine check-up, while the remainder attended for an acute sign or symptom. Sixty-eight per cent (24/35) of the children who were reported to have had a history of toothache subsequently sought dental check-up. There was an association found between maternal dental visits and child dental visits (*p* = 0.021) (Table 2).

### 3.4. Early Feeding Practices

The prevalence of breastfeeding was 84.7%, and among those who were breastfeed the prevalence of exclusive breastfeeding was 13.2%. Also, 67.1% were reported to have often fallen asleep at the breast, 57.8% breast fed six or more times throughout the day and 4.5% were still breast feeding at the time of examination. The mean duration of breast feeding for those who stopped was 16.2 (±11.6) months. Fisher’s exact test showed no association between whether or not a child was breastfed and his/her ECC status (*p* = 0.151) but significant association with S-ECC status (*p* = 0.039).

Only 12.3% were bottle-fed exclusively and of the 86.8% of the children who were bottle-fed, 23.8% of these were still practising bottle feeding at the time of examination. The mean duration of bottle feeding among those who stopped bottle was 27.58 ± 9.6 months. A combination of milk and sweetened cereal (51%) was the most reported bottle content among mothers. This was followed by fruit juices (18.1%) and ‘tea’ (Milo^®^/hot chocolate) (15.1%).

Dietary factors associated with ECC were breast feeding frequency (*p* = 0.048), bottle feeding (*p* = 0.007), sleeping with bottle (*p* ≤ 0.001), added sweetener to bottle contents (*p* = 0.014) and feeding at night or early morning (*p* = 0.031). Dietary factors associated with S-ECC were breast feeding (*p* = 0.039), bottle feeding (*p* = 0.003), sleeping with bottle (*p* ≤ 0.001) and added sweetener to bottle contents (*p* = 0.003) (Table 3).

### 3.5. Snacking Practices

It was found that 87.3% of children were reported to snack between meals and of these 44% had caries. There was no association found between snacking on sweet and savoury snacks between meals and caries experience regardless of the frequency. There was however association between snacking on fruits between meals, where fewer children who snacked on fruit exhibited ECC (*p* = 0.042). This association was also found with S-ECC (*p* = 0.019). There was significant association between consumption of sports drinks and ECC (*p* = 0.004) and also with S-ECC (*p* = 0.004). Frequency of consumption of soft drinks was also significantly associated with S-ECC (*p* = 0.034) but not with ECC (*p* = 0.092). No associations were found either between ECC or S-ECC and children who had pre-chewed food (Table 4).

### 3.6. Oral Hygiene Practices

A small proportion (35.9%) of mothers reported receiving some oral health care advice. When oral hygiene practices were assessed results revealed that the mean age at which tooth brushing was started was 15.31 ± 6.6 months. More than 78.5% of children reportedly brushed two or more times daily and 84.3% were either sometimes or never assisted by a parent/caregiver. (17.5% of children brushed alone and 66.8% were only sometimes assisted.) There was an association with children with ECC and poor oral hygiene (*p* ≤ 0.001). Oral hygiene factors associated with S-ECC were oral hygiene status (*p* ≤ 0.001) and use of kid’s fluoride-free toothpaste (*p* = 0.036). There were no associations between frequency of brushing or type of cleaning agent (adult fluoride toothpaste, kid’s fluoride toothpaste and fluoride-free toothpaste) and ECC (Table 5).

### 3.7. Risk Factors for ECC and S-ECC

Binary logistic regression was applied only to the variables that were associated with ECC and S-ECC. Results showed that the gender of the child (*p* = 0.045) was the only predictor of/risk factor for ECC in this population. On the other hand, none of the factors associated with S-ECC were predictors/risk factors thereof (Table 6).

## 4. Discussion

This study documented a 50% prevalence of caries in a cross section of preschool children in both islands of Trinidad and Tobago. This prevalence is significantly higher than those found in similar studies of preschool children in Trinidad [21] as well as in other Caribbean islands [13,14] and falls on the higher end of reported developing country or international rates [22]. There was also a significant number of children with caries (52.3%) who had severe early childhood caries. The difference between caries experience in boys and girls in this population may be explained by a natural resistance to compliance in boys within this age group and greater emphasis of caregivers in looking after the physical appearance and hygiene of girls.

It was observed that caries prevalence increased with age between 3 and 4 years which mirrored findings in multiple studies [5,23]; however, the noticeable decline between 4 and 5-year-olds in the present study may be most likely due to the low numbers of 5-year-olds who were still in early childhood care rather than in primary school.

Despite the AAPD and BSPD (British Society of Paediatric Dentistry) recommendations of establishing a dental home and a visit by age one year, the mean age of first dental visit at 35 months was consistent with the findings from Naidu et al. Parents appear to generally believe that the ideal time for the child’s first dental check-up should be around the time the child has all primary teeth between ages 3 and 4 years [21,24].

There were also very few children who visited a dental professional for assessment unless associated with an acute dental problem. The small number of children who attended for routine dental check-up as well as those who only sought check-up after some history of toothache seems to reinforce a common practice of parents generally only seeking treatment of primary teeth when a child complains of discomfort or pain, since lay people expect the replacement of temporary teeth by the permanent teeth to be a solution [25]. The need for a child’s earlier dental assessment by a professional should be reinforced and encouraged.

A mother who had previous dental visits was more likely to arrange a dental visit for the child. Unlike many studies that have reported low maternal education levels and low oral health awareness to be associated with ECC, there were no specific maternal factors that influenced caries prevalence in this population [26]. Interestingly, while the findings did not meet the statistical significance set, it should be noted that mothers with university level education was the only category for which a higher percentage of children had no caries compared to the percentage that had caries and may be of clinical significance. A more broad-based approach for oral health education programmes rather than those often targeted to young mothers, those with limited education and limited dental knowledge is therefore recommended.

There were also very few children with ECC who had restorative care (<5%). The high level of caries and number of untreated teeth in this population cannot only be attributed to a general ambivalence toward the role of primary teeth but reported difficulties in accessing available and affordable services for comprehensive dental care of children. This is compounded by the lack of public knowledge about primary prevention due to limited readily available oral health information and education programmes [21,27]. When this is considered, the 10% of children who were deemed ‘caries-free’ but had noncavitated lesions or white spot lesions, without preventive care, would be at risk of rapid progression of dental decay and its consequences.

This study concurs with other studies that have shown that there is a significant association between ECC and bottle-feeding including sleeping with a bottle [28,29,30]. Breast feeding practice and frequency as well as bottle feeding practices may also have roles in S-ECC. In the local context, these findings are also consistent with beliefs that bottle feeding is an integral part of child care in this age group and poses a challenge during weaning as evident by the longer bottle use times [21]. While these bottle feeding practices may be popular, diet modifications addressing these must form a significant part of preventive programmes in this population.

Children with ECC are generally found to have an increased frequency of sugar consumption. This is often associated with sweetened fluids given in the nursing bottle and sweetened solid foods [31]. The frequent use of bottle feeding with contents like milk and sweetened cereals and fruit juices, added sweeteners, as well as frequent consumption of sports and fizzy soft drinks in this population are consistent with the findings that non-milk extrinsic sugars (NMES) have been widely implicated as the cause of caries [32,33] and that the consumption of milk-based formulas for infant feeding, even without sucrose in their formulation, proved to be cariogenic [34]. The findings of this study are consistent with other epidemiological studies that suggest that reasonable fruit intake between meals is not significant in the development of dental caries [35].

The mean age of 15.3 months at which toothbrushing was started in this data set, was much later than the six months or on eruption of first tooth as recommended by the AAPD. The children in this study found with high plaque scores were also found to have more caries experience. This is consistent with other study findings that suggest a positive association between plaque accumulation on primary teeth and ECC risk. [36]. Studies have also shown that increased frequency of toothbrushing and parental involvement can decrease the occurrence of caries lesions on smooth surfaces [4,37,38,39]. This therefore reinforces the need for greater parental engagement at an early age in this population especially given the poor oral hygiene scores and the high number of children reported to be only occasionally assisted during toothbrushing.

Interestingly, there was no association found between the frequency of tooth brushing or the use of a dentifrice (fluoridated or nonfluoridated) and caries experience in this sample. However, there was association found with kid’s nonfluoride toothpaste use and S-ECC. Interpretation of the findings in this study however may be difficult given possible issues of recall bias by the mother and social desirability response bias [4]. The lack of public intervention strategies such as water fluoridation in Trinidad and Tobago is another limiting factor in the prevention of dental decay in this population and reinforces the need for the encouragement of topical measures known to combat the problem.

The study has a number of limitations. The study is a cross-sectional study with small sample size and, therefore, the findings should be generalized with caution in other context. As the criterion for caries diagnosis was based entirely on visual and blunt probe assessment in this study, the result may also be considered an underestimation of the actual caries experience of the sample population, especially for noncavitated proximal lesions that are often detected using dental radiography. Socioeconomic status was not assessed in this study due to a general reluctance of parents to share such information.

## 5. Conclusions

This study revealed that there are high levels of dental caries (early childhood caries and severe ECC) in this population. The caries is associated with various factors such as poor dietary and oral hygiene practices. Gender is a risk factor for this population. Early dental assessment, broad-based oral health education programmes, increased parental/guardian engagement and assistance during oral hygiene practices and greater access to facilities for early childhood caries prevention and management can help alleviate the significant problems of ECC and S-ECC in this population.

## Figures and Tables

**Table 1 dentistry-07-00016-t001:** Demographics, early childhood caries (ECC) and severe ECC (S-ECC).

Demographics	*n* = 342	No Caries *n* = 170	ECC *n* = 172	*p*-Value	S-ECC *n* = 90	Non-S-ECC ^+^ *n* = 252	*p*-Value
Gender							
Male	174 (50.9%)	77 (44.3%)	97 (55.7%)	0.040 **	46 (26.4%)	128(73.6%)	0.959
Female	168 (49.1%)	93 (55.4%)	75 (44.6%)	-	44 (26.2%)	124(73.8%)	-
Ethnicity *							
African	129 (37.7%)	69 (53.5%)	60 (46.5%)	-	31 (24%)	98 (76%)	-
East Indian	74 (21.6%)	32 (43.2%)	42 (56.8%)	-	25 (33.8%)	49 (66.2%)	-
Mixed	129 (37.7%)	65 (50.4%)	64 (49.6%)	0.560	32 (24.8%)	97(75.2%)	0.448
Other	3 (<1%)	2 (66.7%)	1 (33.3%)	-	0 (0%)	3 (100%)	-
Age							
3 years	163 (47.7%)	89 (54.6%)	74 (45.3%)	-	46 (28.2%)	117 (71.8%)	-
4 years	138 (40.4%)	60 (43.5%)	78 (56.5%)	0.154	33 (23.9%)	105 (76.1%)	0.697
5 years	41 (11.9%)	21 (51.2%)	20 (48.8%)	-	11 (26.8%)	30 (73.2%)	-
Region							
North	72 (21.1%)	31 (43.1%)	41 (56.9%)	-	25 (34.7%)	47 (65.3%)	-
Central	82 (24.0%)	44 (53.7%)	38 (46.3%)	-	20 (24.4%)	62 (75.6%)	-
East	61 (17.8%)	36 (59%)	25 (41%)	0.283	8 (13.1%)	53 (86.9%)	0.036 **
South	65 (19.0%)	28 (43.1%)	37 (56.9%)	-	22 (33.8%)	43 (66.2%)	-
Tobago	62 (18.1%)	31 (50%)	31 (50%)	-	15 (24.2%)	47 (75.8%)	-

* cases missing; ** Values were statistically significant (*p* < 0.05); ^+^ Non-S-ECC—all children who did not belong to the S-ECC group.

**Table 2 dentistry-07-00016-t002:** Maternal demographics, ECC, S-ECC and dental exposure.

Maternal Factors	Respondent *n* (%)	No Caries *n* (%)	ECC *n* (%)	*p*-Value	S-ECC *n* (%)	Non-S-ECC *n* (%)	*p*-Value
**Maternal Age ***							
<20	5 (1.5%)	2 (40%)	3 (60%)	-	1 (20%)	4 (80%)	-
20–29	142 (42.4%)	76 (53.5%)	66 (46.5%)	0.388	40 (28.2%)	102 (71.8%)	0.788
30–39	153 (45.7%)	71 (46.4%)	82 (53.6%)	-	36 (23.5%)	117 (76.5%)	-
>40	35 (10.4%)	21 (60%)	14 (40%)	-	8 (22.9%)	27 (77.1%)	-
**Maternal Education ***							
Primary School	30 (8.8%)	12 (40%)	18 (60%)	-	9 (30 %)	21 (70%)	-
Part Secondary	34 (9.9%)	15 (44%)	19 (56%)	0.054	14 (41.2%)	20 (58.8%)	0.169
Secondary	163 (47.7%)	81 (49.7%)	82 (50.3%)	-	43 (26.4%)	120 (73.6%)	-
Vocational training	28 (8.2%)	11 (39.3%)	17 (60.7%)	-	6 (21.4%)	22 (78.6%)	-
University/Tertiary	69 (20.2%)	45 (65.2%)	24 (34.8%)	-	13 (18.8%)	56 (81.2%)	-
**Maternal Dental Visits ***							
No	271 (81.1%)	138 (50.9%)	133 (49.1%)	0.637	66 (24.4%)	205 (75.6%)	0.227
Yes	63 (18.9%)	30 (47.6%)	33 (52.4%)	-	20 (31.7%)	43 (68.3%)	-
**Oral Care Advice ***							
No	211 (64.1%)	111 (52.6%)	100 (47.4%)	0.297	51 (24.2%)	160 (75.8%)	0.211
Yes	118 (35.9%)	55 (46.6%)	63 (53.4%)	-	36 (30.5%)	82 (69.5%)	-
**Child Dental Visits**							
**Maternal Dental Visits**	No	Yes		-	-	-	-
No	194 (71.3%)	78 (28.7%)		-	-	-	-
Yes	34 (54.8%)	28 (45.2%)		0.021 **	-	-	-

* cases missing; ** Values were statistically significant (*p* < 0.05).

**Table 3 dentistry-07-00016-t003:** Early feeding practices, ECC and S-ECC.

Early Feeding Practices *	Response	No Caries *n* (%)	ECC *n* (%)	*p*-Value	S-ECC *n* (%)	Non-S-ECC *n* (%)	*p*-Value
Breast Feeding	No	30 (57.7%)	22 (42.3%)	0.151	8 (15.4%)	44 (84.6%)	0.039 **
	Yes	140 (48.8%)	147 (51.2%)	-	80 (27.9%)	207 (72.1%)	-
Slept at breast	No	32 (53.3%)	28 (46.7%)	0.267	14 (23.3%)	46 (76.7%)	0.250
	Yes	106 (47.7%)	116 (52.3%)	-	64 (28.8%)	158 (71.2%)	-
Still Breast Feeding	No	137 (50.6%)	134 (49.4%)	0.061	73 (26.9%)	198 (73.1%)	0.206
	Yes	4 (26.7%)	11 (73.3%)	-	6 (40%)	9 (60%)	-
Breast Feeding Frequency	<6	50 (56.8%)	38 (43.2%)	0.048 **	22 (25%)	66 (75%)	0.279
	≥6	85 (45.2%)	103 (54.8%)	-	55 (29.3%)	133 (70.7%)	-
Bottle Feeding	No	14 (31.1%)	31 (68.9%)	0.007 **	20 (44.4%)	25 (55.6%)	0.003 **
	Yes	156 (52.9%)	139 (47.1%)	-	69 (23.4%)	226 (76.6%)	-
Slept with Bottle	No	115 (61.2%)	73 (38.8%)	<0.001 **	33 (17.6%)	155 (82.4%)	<0.001 **
	Yes	40 (39.6%)	61 (60.4%)	-	33 (32.7%)	68 (67.3%)	-
Still Bottle Feeding	No	110 (51.6%)	103 (48.4%)	0.352	50 (23.5%)	163 (76.5%)	0.537
	Yes	44 (55%)	36 (45%)	-	19 (23.8%)	61 (76.2%)	
Botte Feeding Frequency	<6	103 (54.5%)	86 (45.5%)	0.478	45 (23.8%)	144 (81.1%)	0.222
	≥6	48 (53.3%)	42 (46.7%)	-	17 (18.9%)	73 (81.1%)	-
Added sweetener to bottle contents	No	117 (59.4%)	80 (40.6%)	0.014 **	37 (18.8%)	160 (81.2%)	0.003 **
	Yes	37 (40.7%)	54 (59.3%)	-	29 (31.9%)	62 (68.1%)	-
Feeding at night/early morning	No	39 (62.9%)	23 (37.1%)	0.031 **	13 (21%)	49 (79%)	0.332
	Yes	120 (47.6%)	132 (52.4%)	-	68 (27%)	184 (73%)	-
Consumed Pre-chewed food	No	141 (49.8%)	142 (50.2%)	0.519	74 (26.1%)	209 (73.9%)	0.544
	Yes	26 (49.1%)	27 (50.9%)	-	14 (26.4%)	39 (73.6%)	-

* Cases missing; ** Values were statistically significant (*p* < 0.05).

**Table 4 dentistry-07-00016-t004:** Dietary practices, ECC and S-ECC.

Dietary Practices *		Response	No Caries *n* (%)	ECC *n* (%)	*p*-Value	S-ECC *n* (%)	Non-S-ECC *n* (%)	*p*-Value
Snack between meals		No	22 (51.2%)	21 (48.8%)	0.854	10 (23.3%)	33 (76.7%)	0.633
		Yes	147 (49.6%)	149 (50.4%)	-	79 (26.7%)	217 (73.3%)	-
Frequency of snacks		<3×/day	100 (52.9%)	89 (47.1%)	0.462	46 (24.3%)	143 (75.7%)	0.201
		>3×/day	29 (44.6%)	36 (55.4%)	-	20 (30.8%)	45 (69.2%)	-
Snack Type	Sweet	No	26 (56.5%)	20 (43.5%)	0.599	10 (21.7%)	36 (78.3%)	0.505
		Yes	116 (51.6%)	109 (48.4%)	-	58 (25.8%)	167 (74.2%)	-
	Savoury	No	55 (53.9%)	47 (46.1%)	0.667	21 (20.6%)	81 (79.4%)	0.213
		Yes	87 (51.2%)	83 (48.8%)	-	48 (28.2%)	122 (71.8%)	-
	Fruit	No	56 (44.8%)	69 (55.2%)	0.042 **	41 (32.8%)	84 (67.2%)	0.019 **
		Yes	86 (58.9%)	60 (41.1%)	-	27 (18.5%)	119 (81.5%)	-
Shared Utensils		No	56 (50.9%)	54 (49.1%)	0.816	25 (22.7%)	85 (77.3%)	0.355
		Yes	112 (49.6%)	114 (50.4%)	-	62 (27.4%)	164 (72.6%)	-
Frequent soft drink intake		No	45 (58.4%)	32 (41.6%)	0.092	13 (16.9%)	64 (83.1%)	0.034 **
		Yes	123 (47.5%)	136 (52.5%)	-	75 (29%)	184 (71%)	-
Frequent sports drink intake		No	138 (54.5%)	115 (45.5%)	0.004 **	58 (22.9%)	195 (77.1%)	0.004 **
		Yes	27 (35.5%)	49 (64.5%)	-	30 (39.5%)	46 (60.5%)	-

* Cases missing; ** Values were statistically significant (*p* < 0.05).

**Table 5 dentistry-07-00016-t005:** Oral hygiene practices, ECC and S-ECC.

Oral Hygiene Practices *	Response	No Caries	ECC	*p*-Value	S-ECC	Non-S-ECC	*p*-Value
Toothbrushing frequency	1×/day	36 (50%)	36 (50%)	-	14 (19.4%)	58 (80.6%)	-
	≥2×/day	132 (50%)	132 (50%)	0.912	73 (27.7%)	191 (72.3%)	0.297
Oral Hygiene Status *							
Good	-	86 (61.9%)	53 (38.1%)	-	21 (15.1%)	118 (84.9%)	-
Poor	-	84 (41.4%)	119 (58.6%)	≤0.001 **	69 (34%)	134 (66%)	≤0.001 **
Cleaning agent *							
Nothing	No	164 (49.8%)	165 (50.2%)	-	85 (25.8%)	244 (74.2%)	-
	Yes	2 (100%)	0 (0%)	0.498	0 (0.0%)	2 (100%)	0.999
Kid’s Fluoride-free toothpaste	No	144 (49.8%)	145 (50.2%)	-	80 (27.7%)	209 (72.3%)	-
	Yes	22 (52.4%)	20 (47.6%)	0.757	5 (11.9%)	37 (88.1%)	0.029 **
Kid’s Fluoride toothpaste	No	37 (50.7%)	36 (49.3%)	-	14 (19.2%)	59 (80.8%)	-
	Yes	129 (50%)	129 (50%)	0.918	71 (27.5%)	187 (72.5%)	0.150
Adult Fluoride toothpaste	No	151 (50.1%)	148 (49%)	-	77 (25.8%)	222 (74.2%)	-
	Yes	15 (46.9%)	17 (53.1%)	0.697	8 (25%)	24 (75%)	0.926
Other (e.g., glycerine)	No	158 (51.3%)	150 (48.7%)	-	78 (25.3%)	230 (74.7%)	-
	Yes	8 (34.8%)	15 (65.2%)	0.126	7 (30.4%)	16 (69.6%)	0.588

* Cases missing; ** Values were statistically significant (*p* < 0.05).

**Table 6 dentistry-07-00016-t006:** Predictors of/risk factors for ECC.

95% CI for OR				
Variable	OR	*p*-Value	Lower	Upper
Gender				
Male	1	-	-	-
Female	11.709	0.045 *	1.625	129.142
Breast feeding frequency				
No	1	-	-	-
Yes	2.471	0.458	0.226	26.982
Feeding night/early morning				
No	1	0.087	0.007	1.145
Yes	0.087	-	-	-
Feeding at night or early morning				
No	1	-	-	-
Yes	0.554	0.125	0.261	1.177
Sleep with Bottle				
No	1	-	-	-
Yes	3.176	0.837	0.091	6.955
Snacks on fruit				
No	1	-	-	-
Yes	3.176	0.310	0.341	29.537
Oral hygiene status				
Poor	1	-	-	-
Good	0.191	0.240	0.012	3.021
Added sweetener to Bottle Contents				
No	1	-	-	-
Yes	0.604	0.683	0.054	6.763

OR—odds ratio. CI—confidence interval. * *p* < 0.05

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
