# Peer review of "Early Childhood Caries in 3 to 5 Year Old Children in Trinidad and Tobago"

_dentistry, 2019, doi:10.3390/dj7010016_

Round 1

Reviewer 1 Report

The manuscript by Percival et al. documents the extent of early childhood caries (ECC) and severe-ECC (S-ECC) in a Caribbean island population.  The manuscript also addresses the behaviors most linked to the presence of ECC and S-ECC and in this respect is largely confirmatory.  However, in this latter realm the manuscript is often misleading by focusing too much on an absolute cut-off for statistical significance.  It is a pet-peeve of this Reviewer that data analyses that yield a p-value of 0.049 are considered to have biological relevance while data with a p-value of 0.051 are more or less dismissed as meaningless.  In a literal sense the manuscript is correct in stating that there was not a statistically significant relationship between the mother’s level of education and a child’s caries experience . . . using the pre-established definition of statistical significance.  But the p-value for these data was 0.054; university-level education was the only category for which a higher percentage of children had no caries compared to the percentage that had caries.  A similar scenario existed for some of the data associated with breastfeeding and caries.  It is also not clear that the early caries children should have been combined with the caries free children.  What would the data have looked like if children with white-spot lesions were included with the caries group?  Overall, the data seemed to be largely confirmatory of other studies but this is sometimes not brought out due to an over emphasis on statistics which should only be considered as a form of evidence, not a proclamation of the value of the entirety of the data. 

Note:  The first sentence of the M&M says that the study area was divided into six regions -- but only five are listed.

Author Response

The authors are very thankful for the comments of the reviewer, which have helped us to improve the paper. We have tried our best to address the concerns raised.

Response to Comments and Suggestions for Authors

** The authors are thankful for the comments of the reviewer which have helped us to improve the paper. We have tried our best to address the concerns.

Sr No.

Comments

Response

1.      

The   manuscript by Percival et al. documents the extent of early childhood caries   (ECC) and severe-ECC (S-ECC) in a Caribbean island population.  The   manuscript also addresses the behaviors most linked to the presence of ECC   and S-ECC and in this respect is largely confirmatory. 

Comments have been well noted

2.      

However,   in this latter realm the manuscript is often misleading by focusing too much   on an absolute cut-off for statistical significance.  It is a pet-peeve   of this Reviewer that data analyses that yield a p-value of 0.049 are   considered to have biological relevance while data with a p-value of 0.051   are more or less dismissed as meaningless. 

Comments have been well noted

3.      

In a   literal sense the manuscript is correct in stating that there was not a   statistically significant relationship between the mother’s level of   education and a child’s caries experience . . . using the pre-established   definition of statistical significance.  But the p-value for these data   was 0.054; university-level education was the only category for which a   higher percentage of children had no caries compared to the percentage that   had caries. 

Comments have been noted and   consideration for possible clinical significance have been included in lines   226-232 and  lines 243 -245

4.      

A similar   scenario existed for some of the data associated with breastfeeding and   caries.  It is also not clear that the early caries children should have   been combined with the caries free children.

Comment about possible clinical   significance of breast feeding has been included in lines 243-244. As the   authors used the WHO dmft index which uses cavitated lesions, the early   caries children were included with caries free children.

5.      

What would   the data have looked like if children with white-spot lesions were included   with the caries group? 

The authors considered this but   thought that it would be more suitable in its current format to be able to   compare findings with similar studies.

6.      

Overall,   the data seemed to be largely confirmatory of other studies but this is   sometimes not brought out due to an over emphasis on statistics which should   only be considered as a form of evidence, not a proclamation of the value of   the entirety of the data. 

Comments have been noted and we   agree with comments

7.      

Note:    The first sentence of the M&M says that the study area was divided into   six regions -- but only five are listed.

The error   has been noted and the correction made in Line 70.

Reviewer 2 Report

Early Childhood Caries in 3 to 5 year old children in Trinidad and Tobago. This manuscript is aimed at evaluating the prevalence and contributory factors of early childhood caries (ECC) and severe ECC (S-ECC) among preschool children of Kindergartens and Early Childhood Centers in Trinidad and Tobago. This manuscript is well written and organized. However, there are some shortcomings that need to be addressed by the authors.

In general:

-       The authors did not follow the STROBE guidelines for reporting cross section studies. Important items are missing in the manuscript. It would be better to be included to improve the quality of this report.

-       English language needs revision and editing specially in the parts presenting statistics. Common words such as “shaking teeth” must be replaced by scientific ones.

Abstract:

-       It would be better if the authors indicate that those islands are in the Caribbean.

-       The authors need to write the index name used in clinical examination.

Materials and methods:

-       The authors did not indicate the dates for sample recruitment.

-       More details are needed on sampling methods used.

Results:

-       In Table 1 percentages for each number in each category such as male, female, African……..etc are needed.

-       3.2. -Prevalence of caries” “3.2-Maternal Factors” “3.3.-” numbers need revision.

-       3.2-Maternal Factors” this subtitle needs revision its following paragraph is not all about maternal factors.

-       Page 4 lines129-130 “The majority of children who visited a dental professional did so for routine check-up (64/107)” it would be better to put percentage as well.

-       Page 5 lines 139-145. “The finding showed that 84.7% of the children were breast fed.” “It was noted that 86.8% of the children were bottle fed” there is a problem with numbers between these two sentences making the total more than 100%.

-       Table 4, there is a missing value in one cell.

-       Table 5, there are cells containing less than 5 items this makes Chi Square test not suitable for them.

-       Regression analysis is needed to identify which factors are associated with ECC in this study.

Conclusion:

-       Conclusion should be based on the results. The authors did not assess preventive programme. These recommendations can be presented in discussion.

Author Response

The authors are very thankful for the comments of the reviewer, which have helped us to improve the paper. We have tried our best to address the concerns raised.

Response to Comments and Suggestions for Authors

Sr   No.

Comments

Response

1.         

Early Childhood Caries in 3 to 5 year old children in   Trinidad and Tobago. This manuscript is aimed at evaluating the   prevalence and contributory factors of early childhood caries (ECC) and   severe ECC (S-ECC) among preschool children of Kindergartens and Early   Childhood Centers in Trinidad and Tobago. This manuscript is well written and   organized. However, there are some shortcomings that need to be addressed by   the authors.

Comments   well noted

2.         

In general:

The authors did not follow the STROBE guidelines for   reporting cross section studies. Important items are missing in the   manuscript. It would be better to be included to improve the quality of this report.

Thank   you for the guidance. The authors have tried to comply with the STROBE   guidelines.

3.         

English language needs revision and   editing specially in the parts presenting statistics. Common words such as   “shaking teeth” must be replaced by scientific ones.

The   term “Shaking teeth” was used in the self- administered questionnaire to   avoid the use of jargon. The authors believe that it has been suitably   addressed in line 143.

4.         

Abstract:

It would be better if the authors   indicate that those islands are in the Caribbean.

Comments   noted and addressed

5.         

The authors need to write the index   name used in clinical examination.

Indices   have been named in lines 98 and 102

6.         

Materials and methods:

The authors did not indicate the   dates for sample recruitment.

Dates   for sample recruitment have been included inline 70

7.         

More details are needed on sampling   methods used.

Details   on sampling methods have been included lines 73-77

8.         

Results:

9.         

In Table 1 percentages for each   number in each category such as male, female, African……..etc are needed.

Percentages   have been included

10.      

3.2. -Prevalence of caries” “3.2-Maternal   Factors” “3.3.-” numbers need revision.

Revisions   completed

11.      

3.2-Maternal   Factors” this subtitle needs revision its following paragraph is not all   about maternal factors.

Comments   noted and title has been revised

12.      

Page 4 lines129-130 “The majority of   children who visited a dental professional did so for routine check-up   (64/107)” it would be better to put percentage as well.

Line   149 – percentage included

13.      

Page   5 lines 139-145. “The finding showed that 84.7% of the children were breast   fed.” “It was noted that 86.8% of the children were bottle fed” there is a   problem with numbers between these two sentences making the total more than   100%.

This   information has been rewritten to reflect the appropriate numbers in lines   158-165.

14.      

Table 4, there is a missing value in   one cell.

Omission   noted and value added

15.      

Table   5, there are cells containing less than 5 items this makes Chi Square test   not suitable for them.

Yates   correction for Chi-Square has been applied to cells with numbers less than 5

16.      

Regression analysis is needed to   identify which factors are associated with ECC in this study.

The   authors appreciate the comment from the reviewer about using logistic   regression analysis however given the nature of the data the authors chose to   use chi square rather than multiple regression logistic.

17.      

Conclusion:

Conclusion should be based on the results. The authors did   not assess preventive programme. These recommendations can be presented in   discussion.

Noted   and conclusions revised.

Reviewer 3 Report

Introduction: 

1.      You describe ECC and SECC and define the problem that you are trying to address with your research.  You have gathered an extensive amount of information as part of your questionnaire that will prove to be useful in your research.  You also clearly lay out your aims for the study. 

2.      It may be helpful to describe the population makeup of Trinidad and Tobago and why the caries rate is high in the English speaking Caribbean (from 21-29%).  How does this compare to other island nations  and neighboring Venezuela?

Materials and Methods:

1.      In the first paragraph (P2 line 62-65) you mention that you divided Trinidad Tobago into 6 regions but you do not explain why or if this truly represents the demographics of Trinidad Tobago? With the kindergartens and early childhood centers that were chosen, do these accurately represent the SES and demographics of Trinidad and Tobago? It may be beneficial to explain in more detail that the random sampling truly represents the population ofTrinidad and Tobago.

2.      On p2 line 75-81, you mention the questionnaire, it would be helpful to have a figure with the questionnaire or have it as an appendix document. It is too vague as to what was asked in the questionnaire.

Results:

1.      Page 3 line 95, you discuss inter-examiner calibration, however, was there any intra-examiner calibration.  Was there a portion of the children screened twice by the same person to see if they agreed with themselves?

2.      Page 3 line 101, please state your response rate.

3.      Page 3 line 102-105 you describe the ethnicity of your subject population. Does this match with the makeup of the general population of Trinidad and Tobago?

4.      Your tables provide a lot of information that is helpful to describe your findings.  In table 1 you have a column for S-ECC but in subsequent tables you do not examine S-ECC. This could strengthen your paper.

5.      Page 5 line 139-149 you discuss feeding practices and these are shown in table 3.  You mention that 84.7% of children were breastfed and that 86.8% were bottle-fed. There appears to be overlap of children being breastfed and bottle-fed, but there is not mention in the results of this overlap.  This could be a category in Table 3 as the “bottle feeding” category and “Breast-fed” category are not purely one or the other, but a mix of both. I would recommend that this analysis be completed to more accurately describe your results.

6.      In table 4, under “Frequent sports drink intake” there is missing data for those who do not drink sports drinks but have caries.

Discussion:

1.      Overall your discussion is thorough and you compare your results to other studies and comments on limitations to the current study.

2.      In your results, you mention that males had significantly more caries when compared to females, but you do not elaborate on why this might be.  (Page 3 Line 16-117)

3.      Are there any issues, such as access to care/number of dentist, insurance, cost, government programs, etc. that may be contributing to the >50% caries rate, mostly untreated dental decay?  If so, please elaborate in your discussion.

Thank you for your submission.

Author Response

The authors are very thankful for the comments of the reviewer, which have helped us to improve the paper. We have tried our best to address the concerns raised.

Response to Comments and Suggestions for Authors

Sr No.

Comments

Response

1.       

Introduction: 

You describe ECC and SECC and define the   problem that you are trying to address with your research.  You have   gathered an extensive amount of information as part of your questionnaire   that will prove to be useful in your research.  You also clearly lay out   your aims for the study. 

Comments are well noted.

2.       

It may be helpful to describe the population   makeup of Trinidad and Tobago and why the caries rate is high in the English   speaking Caribbean (from 21-29%).  How does this compare to other island   nations and neighboring Venezuela?

Comments noted and information   updated in lines 47-51.

There is no data available for   neighboring Venezuela

3.       

Materials and Methods:

In the first paragraph   (P2 line 62-65) you mention that you divided Trinidad Tobago into 6 regions   but you do not explain why or if this truly represents the demographics of   Trinidad Tobago? With the kindergartens and early childhood centers that were   chosen, do these accurately represent the SES and demographics of Trinidad   and Tobago? It may be beneficial to explain in more detail that the random   sampling truly represents the population of Trinidad and Tobago.

Error noted and correction has   been made. Information about demographics and representation have been   included in lines 52-57, 69-77.

4.       

On p2 line 75-81, you   mention the questionnaire, it would be helpful to have a figure with the   questionnaire or have it as an appendix document. It is too vague as to what   was asked in the questionnaire.

The authors noted that   questionnaires are not usually included with papers. Access to the   questionnaire is available if needed.

5.       

Results:

Page 3 line 95, you discuss inter-examiner   calibration, however, was there any intra-examiner calibration.  Was   there a portion of the children screened twice by the same person to see if   they agreed with themselves?

Intra and inter- examiner   calibration information has been included line 109-112.

6.       

Page 3 line 101, please state your response   rate.

Line 117- response rate added

7.       

Page 3 line 102-105   you describe the ethnicity of your subject population. Does this match with   the makeup of the general population of Trinidad and Tobago?

Comments noted and information   included in lines 52 -57, 120-123

8.       

Your tables provide a lot of information that   is helpful to describe your findings.  In table 1 you have a column for   S-ECC but in subsequent tables you do not examine S-ECC. This could   strengthen your paper.

Comments noted and additional   information regarding S-ECC has been added to all tables.

9.       

Page 5 line 139-149   you discuss feeding practices and these are shown in table 3.  You   mention that 84.7% of children were breastfed and that 86.8% were bottle-fed.   There appears to be overlap of children being breastfed and bottle-fed, but   there is not mention in the results of this overlap.  This could be a   category in Table 3 as the “bottle feeding” category and “Breast-fed”   category are not purely one or the other, but a mix of both. I would   recommend that this analysis be completed to more accurately describe your   results.

This information has been   rewritten to reflect the appropriate numbers.

( Lines 158-165)

10.    

In table 4, under “Frequent sports drink   intake” there is missing data for those who do not drink sports drinks but   have caries.

Omission noted and value added

11.    

Discussion:

Overall your   discussion is thorough and you compare your results to other studies and   comments on limitations to the current study.

Comment noted

12.    

In your results, you mention that males had   significantly more caries when compared to females, but you do not elaborate   on why this might be.  (Page 3 Line 16-117)

Comment noted and addressed in   lines 206-209

13.    

Are there any issues, such as access to   care/number of dentist, insurance, cost, government programs, etc. that may   be contributing to the >50% caries rate, mostly untreated dental   decay?  If so, please elaborate in your discussion.

The authors believe that these   issues have been addressed in lines 235-240

Round 2

Reviewer 2 Report

The manuscript is much improved, however for better quality regression analysis in addition to Chi square would improve the results to identify which factors are associated with early childhood caries in this population. 

Author Response

Thank you for your comments. Logistical regression has been conducted and the methods, results and conclusions have been reviewed and revised.

Reviewer 3 Report

Thank you for your re-submission.

Only a few minor comments:

In tables 1, 2, 4, AND 5 you use the "caries" as a heading (no caries and caries columns), however, in table 3 instead of "caries" you use ECC.  I would suggest using one label for all tables.

In tables 1-5 you use the term non-S-ECC, does that mean ECC?  Currently non-S-ECC is not defined.  If it means ECC it might be better to place that as a label for that column. If you would like to use non-S-ECC please place a note in the footnote of the table with a brief definition.

For those who exclusively breast fed (13.2%), where did those children fall in terms of ECC and S-ECC?  In the discussion (lines 261-263) you mention that this study supports that breast feeding between 13-23 months has no effect on dental caries and that breast feeding has clinical significance in S-ECC development. Your reported p value of 0.059, which is trending to show that people who breast feed have fewer S-ECC.  However this is still not purely breast fed only since 71.5% both bottle and breastfed and can be misleading - which feeding habit is influencing the results? Roughly 45 people in the breastfeeding group, breastfed exclusively.  It would be clearer evidence to support this conclusion if you report the data from these 45 people in table 3. Are there subject who bottle fed exclusively? If so this could be another pure group.  Then the subjects who both bottle fed and breastfed could be another group.  If you can show that the 45 people who breastfed only have trending or sig. less S-ECC then this could lend itself to support that the mixed groups lower S-ECC compared to non S-ECC might be influenced by breastfeeding. This would strengthen your conclusion and paper.  

Author Response

In tables 1, 2, 4, AND 5 you use the "caries" as a heading (no caries and caries columns), however, in table 3 instead of "caries" you use ECC.  I would suggest using one label for all tables.

Response: Thank you for your comments. The inconsistency has been noted and the changes made to the heading (No caries and ECC)

In tables 1-5 you use the term non-S-ECC, does that mean ECC?  Currently non-S-ECC is not defined.  If it means ECC it might be better to place that as a label for that column. If you would like to use non-S-ECC please place a note in the footnote of the table with a brief definition.

Response: A footnote has been placed at the end of Table 1 "non S-ECC - all children who do not belong to the S-ECC group ."  (This group includes ECC as well as caries free children.)

For those who exclusively breast fed (13.2%), where did those children fall in terms of ECC and S-ECC?  In the discussion (lines 261-263) you mention that this study supports that breast feeding between 13-23 months has no effect on dental caries and that breast feeding has clinical significance in S-ECC development. Your reported p value of 0.059, which is trending to show that people who breast feed have fewer S-ECC.  However this is still not purely breast fed only since 71.5% both bottle and breastfed and can be misleading - which feeding habit is influencing the results? Roughly 45 people in the breastfeeding group, breastfed exclusively.  It would be clearer evidence to support this conclusion if you report the data from these 45 people in table 3. Are there subject who bottle fed exclusively? If so this could be another pure group.  Then the subjects who both bottle fed and breastfed could be another group.  If you can show that the 45 people who breastfed only have trending or sig. less S-ECC then this could lend itself to support that the mixed groups lower S-ECC compared to non S-ECC might be influenced by breastfeeding. This would strengthen your conclusion and paper. 

Response: The authors acknowlege that the statement was misleading. Fisher test of equality of two proportions and logistic regression were conducted to determine the associations and anyrespective risk factors. The discussion has been modified accordingly.

Round 3

Reviewer 2 Report

No further comments 

Reviewer 3 Report

Thank you for the time and effort that you have placed in this study.